# Potential future exposure of European land transport infrastructure to rainfall-induced landslides throughout the 21[st] century

Matthias Schlögl[1,2] and Christoph Matulla[3]

[1]Transportation Infrastructure Technologies, Austrian Institute of Technology (AIT), Vienna, Austria
[2]Institute of Applied Statistics and Computing, University of Natural Resources and Life Sciences (BOKU), Vienna, Austria
[3]Department for Climate Research, Zentralanstalt für Meteorologie und Geodynamik (ZAMG), Vienna, Austria

*Correspondence to:* Matthias Schlögl (matthias.schloegl@ait.ac.at), Christoph Matulla (christoph.matulla@zamg.ac.at)

**Abstract.** In the face of climate change, the assessment of land transport infrastructure exposure towards adverse climate events is of major importance for Europe's economic prosperity and social wellbeing. In this study, a climate index picturing rainfall patterns which trigger landslides in Central Europe is analyzed until the end of this century and compared to present day conditions. The analysis of potential future developments is based on an ensemble of dynamically downscaled climate projections which are driven by the SRES A1B socio-economic scenario. Resulting regional scale climate change projections across Central Europe are concatenated with Europe's road and railway network. Results indicate overall increases of landslide occurrences. While flat terrain at low altitudes exhibits an increase of about one more potentially landslide-inducing rainfall period per year until the end of this century, higher elevated regions are more affected and show increases of up to 14 additional periods. This general spatial distribution emerges already in the near future (2021-2050) but gets more pronounced in the remote future (2071-2100). Since largest increases are to be found in the Alsace, potential impacts of an increasing amount of landslides are discussed using the example of a case study covering the Black Forest mountain range in Baden-Württemberg by further enriching the climate information with and additional geodata. Derived findings are suitable to support political decision-makers and European authorities in transport, freight and logistics by offering detailed information on which parts of Europe's land-bound transport network are at particularly high risk concerning landslide activity.

## 1 Introduction

Given the outstanding importance of land transport modes for Europe's social and economic prosperity, the free and uninterrupted movement of persons and freight is of central magnitude. For instance, the accessibility of healthcare facilities, the supply of daily goods as well as a broad range of services to communities rely on the continuous availability of roads and railway connections.

Extensive soil sealing across Europe (Nestroy, 2006), climate change (European Environment Agency, 2014; Loveridge et al., 2010) and extreme weather impacts (Schlögl and Laaha, 2017) challenge the resilience of transport systems, which have thus grown into a matter of major concern – not only because of physical damages to assets (Kellermann et al., 2015), but also due to potential overall societal losses caused by network failures and interruptions, which often exceed infrastructure damages by far (Postance et al., 2017; Pfurtscheller and Vetter, 2015; Bíl et al., 2015; Pfurtscheller, 2014; Pfurtscheller and Thieken,

2013; Meyer et al., 2013). Thus, the assessment of land transport infrastructure exposure towards adverse climate events and related natural hazards is of great importance for Europe's economy, for its intermodal transport, its freight and logistics networks as well as for settlements in hazard-prone regions (Koetse and Rietveld, 2009; Doll et al., 2014). Therefore, information on current climate and its variability as well as on potential future climate changes is of prime importance for proactive planning and the development of adaptation strategies concerning operation, maintenance, reinforcement and construction works and for civil protection.

Climate change and alterations in extreme weather events – which are affecting ecosystems and man-made infrastructure – have been investigated since some decades by now (e.g. IPCC, 1990, 1995, 2001, 2007, 2012, 2014b). Among various menacing hazards, landslides stand out as destructive hazards to the functional effectivity and structural integrity of land-bound transport systems, since they cause long-lasting downtimes and exceedingly expensive repair works. Even though observed changes in extreme events in terms of frequency, intensity and duration may not be directly associated with global warming, trends concerning landslide occurrences are visible (Gariano and Guzzetti, 2016; McBean, 2011; Crozier, 2010). Therefore, information about expected future changes in landslide occurrence and its impacts on land transport infrastructure may provide a basis for the implementation of adequate adaptation measures.

Ever since the impact of mankind on the climate system has been proven (e.g. IPCC, 2000) it has become increasingly important to derive estimates of future climate states. Global Climate Models are presently the state-of-the-art tools to investigate future climate developments, mimicking global scale consequences for the climate system of the Earth. GCM results are valid on global and continental scales and have to be 'downscaled' (e.g. von Storch et al., 1993) to regional scales for the assessment of future hazard occurrences. This approach has been shown to yield valuable results for the evaluation of rainfall-induced landslides at regional scales (Gariano et al., 2017a; Matulla et al., 2017).

Studies about the effects of climate change on landslide activity have gained center stage in recent years. Crozier (2010) was one of the first to systematically examine the underlying mechanisms linking climate change impacts and slope stability. It was pointed out in this work that while there is a strong theoretical basis for increased landslide activity as a result of a changing climate, a certain extent of uncertainty remains due to inherent incompleteness and inhomogeneities of historic data on climate and landslide recordings, the nature of scenario-based projections and the lack of downscaled data at an appropriate spatial resolution. However, changes in annual and seasonal precipitation patterns (in terms of both precipitation totals and intensity) have been detected as key determinants affecting landslides (Gariano and Guzzetti, 2016; Sidle and Ochiai, 2013).

Since the difficulties of establishing a universal relationship between climate change and landslides across the entire of Europe was pointed out by Dikau and Schrott (1999), citing the complexity of the problem as the main obstacle, several authors have undertaken efforts to establish such relationships for different parts of the continent (e.g. Gariano et al., 2017b; Gariano and Guzzetti, 2016; Jaedicke et al., 2014; Promper et al., 2014; Van Den Eeckhaut et al., 2012; Crozier, 2010).

The derivation of climate change induced future hazards is essentially based on two key components: (i) on sets (so-called ensembles) of Global Climate Model (GCM) runs, which are driven by potential future pathways of mankind and cascaded down to regional-scales via downscaling techniques (e.g. von Storch et al., 1993; Matulla et al., 2003) and (ii) on relationships (so-called climate indices – CIs) between regional-scale climate phenomena (e.g. long term rain exceeding certain thresholds)

and damage events. Ensembles of climate change projections depicting corridors of future climate evolutions and CIs can be arranged in so-called Cause-Effect Tensors, which have already been successfully applied to access potential future damage events to European transport infrastructure (Matulla et al., 2017).

Rainfall periods exceeding certain thresholds have been found to serve as a proper proxy for landslide occurrences and been applied in Central Europe (Peruccacci et al., 2017; Matulla et al., 2017; Gariano et al., 2017a; Brunetti et al., 2010; Guzzetti et al., 2008, 2007; Dixon and Brook, 2007). Based on these findings, we employ a CI that inherits intensity and totals of severe rainfall events, which have been shown to act as a primary trigger of rapid-moving landslides in Central Europe (Gariano and Guzzetti, 2016).

It has to be noted that there are no unambiguous criteria that could be used for a geographically distinct delimitation of the Central European region. In this study, we adjust the most widely accepted definition of Central Europe – as used e.g. in the World Factbook (CIA, 2017) – according to the given data availability. Thus, the following countries are considered (at least partially) in the present study: Austria, Belgium, Czech Republic, France, Germany, Liechtenstein, Luxembourg, the Netherlands and Switzerland.

In order to illustrate the implications of these meteorological impacts in a more practical context, results of the CI exposure mapping are enriched with additional environmental information at the example of a selected, risk-prone area located around the tripoint of France, Germany and Switzerland in the Upper Rhine valley. Hence, we amplify the scope of the discussion by embedding mesoscale CI data (obtained via downscaling of GCMs) into a specific regional context featuring additional information about environmental properties such as data about geomorphology, soil or land cover.

This study is devoted to the assessment of climate change driven landslide hazards to European transport infrastructure (rails and roads) in the near (2021-2050) as well as the remote (2071-2100) future. Results are based on the so-called A1B socio-economic scenario (IPCC, 2000) and shall provide European Transport Authorities with auxiliary information for setting up cost effective and spatially targeted protection measures in order to safeguard Europe's transport system in the future.

## 2 Data

### 2.1 Climate

Since the goal of this study is to investigate potential changes in landslide occurrence jeopardizing Europe's transport infrastructure, climate data used refer to two future periods relative to past conditions: the near future (2021–2050) and the remote future (2071–2100).

The assessment of associated future climate threats jeopardizing Central European transport assets relies on ensembles of daily-based, regional-scale climate change projections. Such ensembles of climate change projections – driven by potential pathways of mankind – can be generally derived via two techniques: dynamical and statistical-empirical downscaling (Matulla et al., 2002; von Storch et al., 1993). Here we make use of an ensemble consisting of 17 climate model runs that has been applied to create dynamically downscaled and bias-corrected (and hence physically consistent) future projections of climate states for climate change impact investigations (Imbery et al., 2013). The projections are driven by the so-called A1B SRES

socio-economic scenario, which describes a future world characterized by a dynamical economic development, decreasing social and income inequalities a rapid dissemination of new and efficient technologies as well as a balanced use of energy sources (IPCC, 2000).

## 2.2 Infrastructure

The graph of the road infrastructure network is based on a data extract of OpenStreetMap (OSM) obtained in June 2017. The high-level road network used in this study is derived from the OSM data set by applying a filter selecting only the following highway tags: `motorway`, `motorway_link`, `trunk`, `trunk_link`, `primary` and `primary_link`. The network selected in this way contains highways as well as major roads(OSM, 2017). The railway network is represented by the Natural Earth Railroads vector data set, version 3.0.0 (Natural Earth, 2017). In order to obtain results at an adequate resolution,

all connections exceeding 500 m have been split into multiple segments every 500 m.

## 2.3 Topography

The EU-DEM v1.1 Digital Elevation Model (DEM) is used in this study to analyze topographic properties. This DEM, which is a hybrid product based on SRTM (Shuttle Radar Topography Mission) and ASTER (Advanced Spaceborne Thermal Emission and Reflection Radiometer) DEM data, fused by a weighted averaging approach, is provided by the Copernicus programme

(Copernicus, 2017). The slope and Terrain Ruggedness Index (TRI) are derived from this DEM with `gdaldem` using Horn's formula (Horn, 1981).

## 2.4 Additional geodata

In addition, data sets about geological and soil properties as well as data about rainfall erosivity and land cover data have been used to augment the discussion of the implications imposed by the derived CI. CORINE Land Cover data are provided by

the Copernicus programme (Copernicus, 2017). The geology data set (IGME 5000) is provided by the Federal Institute for Geosciences and Natural Resources (Asch, 2005), while soil data and rainfall erosivity are accessible through the European Soil Data Centre (Panagos et al., 2012). Please note that all of these data sources are freely accessible. The respective sources for all data sets are referenced in the *data availability* section at the end of the paper.

## 3   Methods

The gridded data sets of daily precipitation totals derived from the 17-member ensemble of climate model runs forced by the A1B socio-economic scenario are available on a 5 km grid across large parts of Central Europe (c.f. Fig. 1). These raster data sets were used to calculate the landslide CI for Central Europe at the same spatial resolution. The CI for potentially landslide triggering events was established by using a proxy indicating precipitation periods extending over at least three days, while generating overall totals of more than 37.3 mm, and exhibiting at least one day with a total exceeding 25.6 mm. This threshold

for landslide activity was proposed by Guzzetti et al. (2008) and was recently used to establish a CI for landslide occurrence

based on threshold exceedance in Matulla et al. (2017). Changes in hazard occurrences (predefined via this CI) over time can be analyzed by comparisons of past and potential future probability density functions.

In order to obtain infrastructure exposure of the road and railway assets in Central Europe at the network level, CI values from above described gridded data sets were mapped to the underlying road and rail segments. This was achieved by extracting and interpolating corresponding values from the raster grid towards the road and railway network graphs.

For the sake of putting the obtained landslide exposure maps into a more expressive context concerning the interpretation and discussion of practical implications, maps depicting slope angles, terrain ruggedness, geomorphologic properties as well as rainfall erosivity and land cover were created for a particularly risk-prone area, henceforth called "target region". The obtained landslide exposure, which is based on the selected CI is then analysed and discussed against the background of these additional environmental features that are widely used in landslide studies (e.g. Jaedicke et al., 2014; Papathoma-Köhle and Glade, 2013; Van Den Eeckhaut et al., 2012).

As far as topography is concerned, the slope angle and the TRI are considered. Slope angles are known to be a key parameter in estimating susceptibility to developing earth flows (Donnarumma et al., 2013). The TRI is defined as the mean difference between a central pixel and its surrounding cells (Wilson et al., 2007) and can be used to quantify landscape heterogeneities, which could exert influence on the localization of the triggering area of shallow landslides (Persichillo et al., 2016). Both properties are derived from the DEM using Horn's formula (Horn, 1981).

Information concerning the nature and the properties of the ground, which are known to be important aspects affecting landslide susceptibility, rock and soil type have been mapped for the target region. Lithological types are obtained from the International Geological Map of Europe and Adjacent Areas (Asch, 2005), and the dominant Soil Typology Unit is mapped according to the World Reference Base for Soil Resources as available in the Euopean Soil Database (Panagos et al., 2012; Panagos, 2006).

The data sets about topographic, geological and soil properties as well as data about rainfall erosivity and land cover data have solely been used to augment the discussion of the implications imposed by the derived CI.

## 4 Results and Discussion

In terms of the time horizon under consideration, results of this study refer to two different time periods throughout the twenty-first century. The first period (near future) covers the years 2021 to 2050, while the second period (remote future) ranges from 2071 to 2100. In this context it has to be noted that projected precipitation values obtained from the ensemble of 17 climate model runs are relative to current climate conditions. This entails that all results have to be interpreted as variations that are averaged over the whole future period with respect to present day conditions as a baseline reference.

### 4.1 Central Europe

Results are visualized as exposure maps for both the high-level road network (Fig. 1) as well as the railway network (Fig. 2) in Central Europe. In order to provide information about future probability density functions of the selected CI (and not only

about the best estimate), their quartiles (i.e. 25th percentile, median and 75th percentile) are reported for each of the two time periods under consideration, resulting in six different facets.

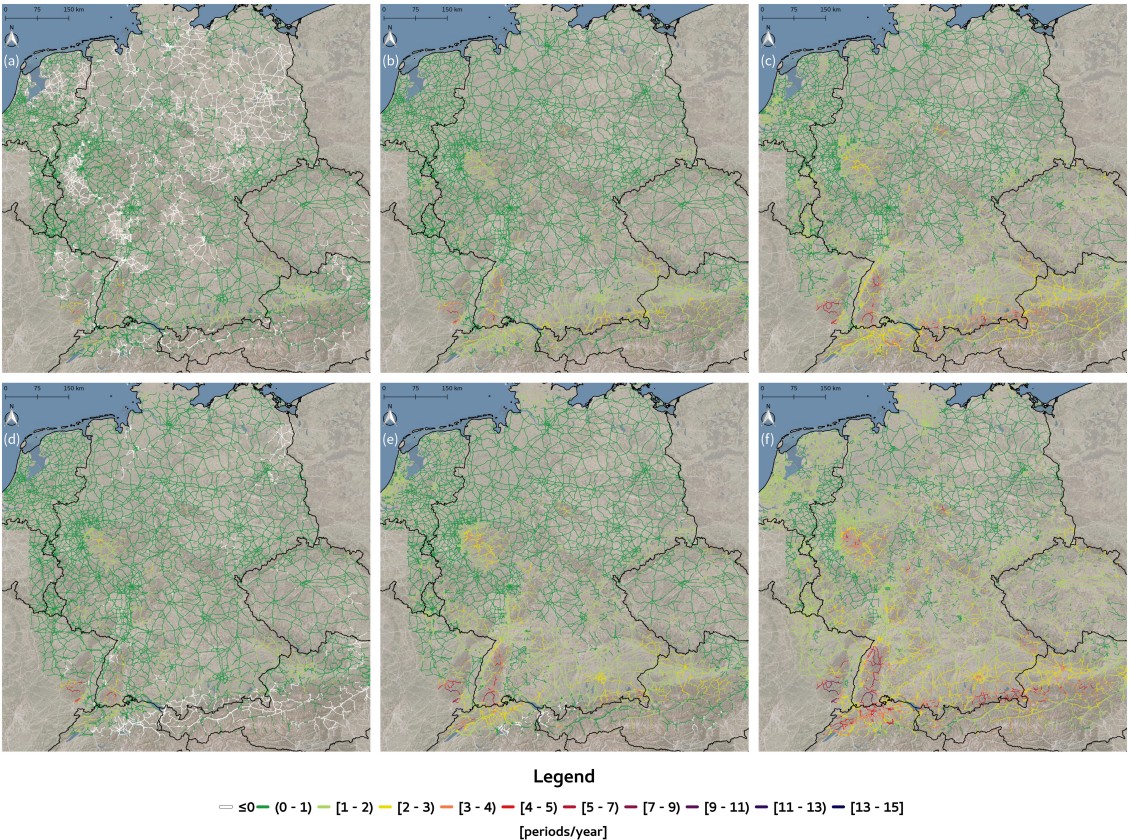

**Legend**

⊘ ≤0 — (0 - 1) — [1 - 2] — [2 - 3] — [3 - 4] — [4 - 5] — [5 - 7] — [7 - 9] — [9 - 11] — [11 - 13] — [13 - 15]

[periods/year]

**Figure 1.** Projected changes in the annual number of periods exceeding rainfall thresholds for the possible occurrence of landslides (RR > 25.6 $\mathrm{mm/d}$ and RR > 37.3 $\mathrm{mm/3d}$) concerning the high-level road network in Central Europe. The first row of each figure (a –c) refers to the near future, the second (d –f) row displays projection results for the remote future. The three columns represent the quartiles in increasing order respectively, with (a) and (d) displaying the lower quartile, (b) and (e) displaying the median, and (c) and (f) displaying the upper quartile

Generally speaking, results show that the most risk-prone areas are located in regions that are characterized by structured topography, e.g. in uplands or in the Alpine forelands. Concerning near future changes, at least a slight increase of rainfall-induced landslides has to be expected all over the entire region. As expected, only the near future 25th percentile does not show any changes in potential landslide occurrences in certain lowland areas, which are mainly located in north-eastern and middle Germany. This is consistent with the physics of maritime influenced regions since large water bodies tend to damp rapid changes in surrounding areas. In contrast, there are several regions that are expected to face up to seven additional landslide-inducing periods, namely the Vosges, the Black Forest ("Schwarzwald"), the Swabian Jura ("Schäbische Alb"), the Bergisches

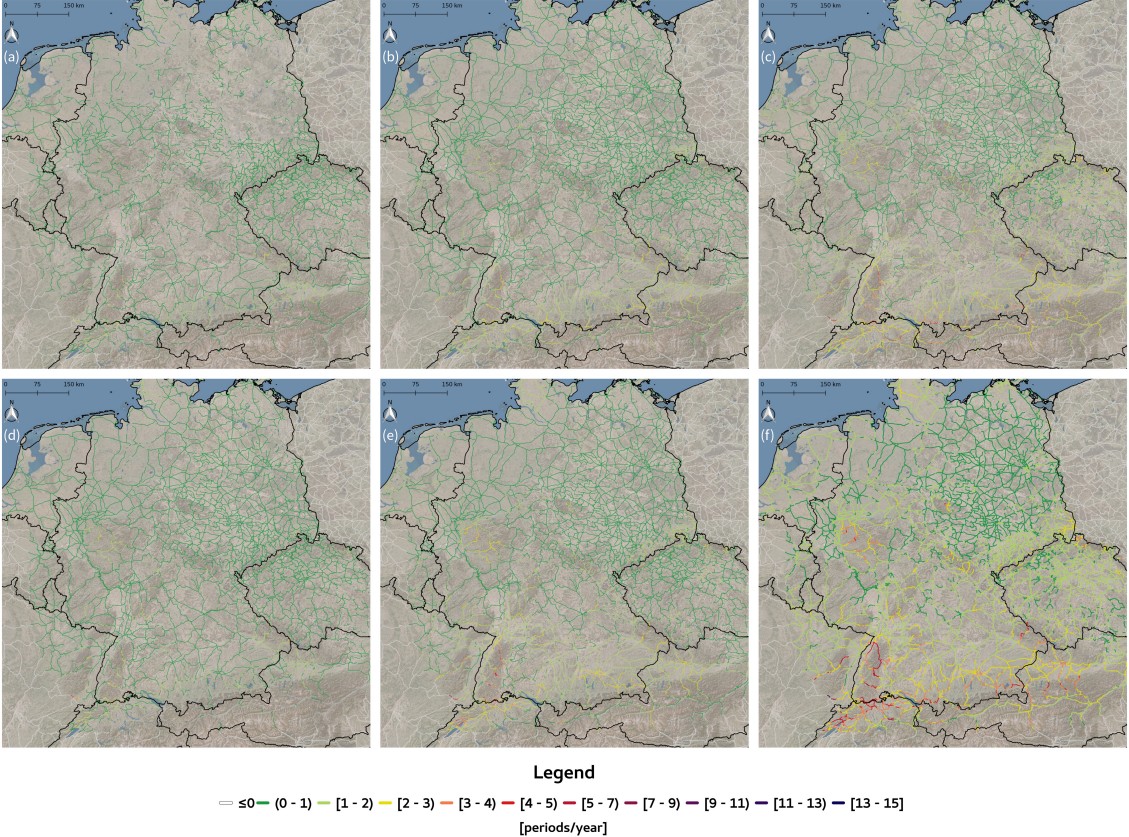

**Legend**

⎯ ≤0 ⎯ (0 - 1) ⎯ [1 - 2] ⎯ [2 - 3] ⎯ [3 - 4] ⎯ [4 - 5] ⎯ [5 - 7] ⎯ [7 - 9] ⎯ [9 - 11] ⎯ [11 - 13] ⎯ [13 - 15]
[periods/year]

**Figure 2.** Projected changes in the annual number of periods exceeding rainfall thresholds for the possible occurrence of landslides (RR > 25.6 mm/d and RR > 37.3 mm/3d) concerning the railway network in Central Europe. The first row of each figure refers to the near future, the second row displays projection results for the remote future. The three columns represent the quartiles in increasing order respectively, with (a) and (d) displaying the lower quartile, (b) and (e) displaying the median, and (c) and (f) displaying the upper quartile

Land, the Jura Mountains, the foothills of the Northern Limestone Alps, the Alpine foreland in Austria and Bavaria as well as the Bohemian Forest ("Šumava" or "Böhmerwald").

Our findings show that changes in occurrence frequencies of landslide-triggering extreme climate events slightly increase over time, as is clearly visible when comparing the second row of Figs. 1 and 2 to the first row of each Figure respectively. Basically, the same patterns apply, but the magnitude of the changes is increased in the remote future period. This is largely consistent with findings from the IPCC's fifth assessment report (IPCC, 2014a). It has to be noted, though, that the overall occurrence of landslide-inducing rainfall events appears to increase only slightly in pace throughout the twenty-first century. Nevertheless, the aforementioned areas along the north side of the Alps are likely to experience substantially increased landslide activity in the remote future compared to current climate conditions, pointing to an acceleration of landslide occurrences and showing a possible increase of up to 14 additional landslide-triggering rainfall events (c.f. Tab. 1).

**Table 1.** Summary statistics for the six different facets displayed in Figs. 1 and 2 based on all raster cells covering road or rail infrastructure links. Results refer to the relative changes compared to present-day-conditions, values indicate the magnitude of changes in the number of periods exceeding the CI thresholds.

| Period | Minimum | First quartile | Median | Third quartile | Maximum |
|---|---|---|---|---|---|
| near future (first quartile) | -1.53 | 0.00 | 0.13 | 0.33 | 4.97 |
| near future (median) | -0.07 | 0.30 | 0.50 | 0.83 | 5.97 |
| near future (third quartile) | 0.13 | 0.63 | 0.90 | 1.40 | 7.37 |
| remote future (first quartile) | -6.08 | 0.17 | 0.33 | 0.57 | 6.72 |
| remote future (median) | -1.80 | 0.53 | 0.73 | 1.13 | 10.54 |
| remote future (third quartile) | -0.53 | 0.90 | 1.20 | 1.83 | 13.97 |

As uncertainty about future projections increases, the spread between the first and third quartile of the projections grows as well, showing variations of up to seven events for the far future period. This coincides with our expectations, since coherence amongst projections decreases towards the end of this century. In addition, the underlying patterns of the geographical distributions of the results are similar, indicating overall robust results.

5     With respect to infrastructure exposure towards potential landslide susceptibility the following regions can be discriminated:

1. regions with a substantial increase in rainfall-induced landslide activity are the Jura Mountains, the Vosges, the Black Forest, the Swabian Jura, the Bavarian Prealps, the foothills of the Austrian Alps and the Bohemian Forest;

2. regions with less pronounced variations in the CI under consideration, that are nonetheless clearly distinct from their surroundings are the Bergisches Land, the Harz, the Fichtel Mountains, Vogtland and the Ore Mountains;

10    3. the rest of Central Europe, where changes in occurrence frequencies of rainfall-induced landslides have to be expected only to a minor extent.

As far as the backbones of the European road and railway infrastructure – the so-called Trans-European Transport Networks (TEN-T) – are concerned, most TEN-T core-network corridors are likely to be affected by increased landslide activity. In particular, the Rhine-Danube corridor (Strassbourg – Karlsruhe, Munich – Rosenheim – Salzburg – Linz, Regensburg – Passau – Linz), the Scandinavian-Mediterranean (Munich – Rosenheim – Kufstein), the Rhine-Alpine and North Sea-Mediterranean corridors (Karlsruhe – Strasbourg – Basel, onwards to Belfort, Bern, Lucerne and Zurich) as well as the North-Sea Baltic corridor (in the area of Wuppertal) have been identified to face an increased exposure to landslide inducing rainfall events.

## 4.2 Target area

Meteorological impacts on geomorphological events and driving landscape-change processes over short time scales have been found to have serious consequences, particularly in climatically sensitive regions such as the European Alps Keiler et al. (2010).

Yet it has to be noted that while this analysis refers to potentially hazardous rainfall events that may trigger landslides, other environmental information have not been taken into account so far. Therefore, the consideration of reduced static information commonly used for landslide susceptibility evaluations (Günther et al., 2014, 2013; Hervás et al., 2007) is illustrated and discussed at the example of a particularly risk-prone area.

5      The selected region is centered around the lowlands of the Alsace and the Black Forest mountain range, covering parts of France, Germany, Switzerland, Luxembourg and Austria (Fig. 3). This area has been selected for two reasons. First, it is one of the regions showing the largest increase in the selected CI. Second, consequences of interruptions are quite severe in this area, as the recent mass movements in the Rhine Valley between Baden-Baden and Rastatt have shown (Ackeret, 2017).

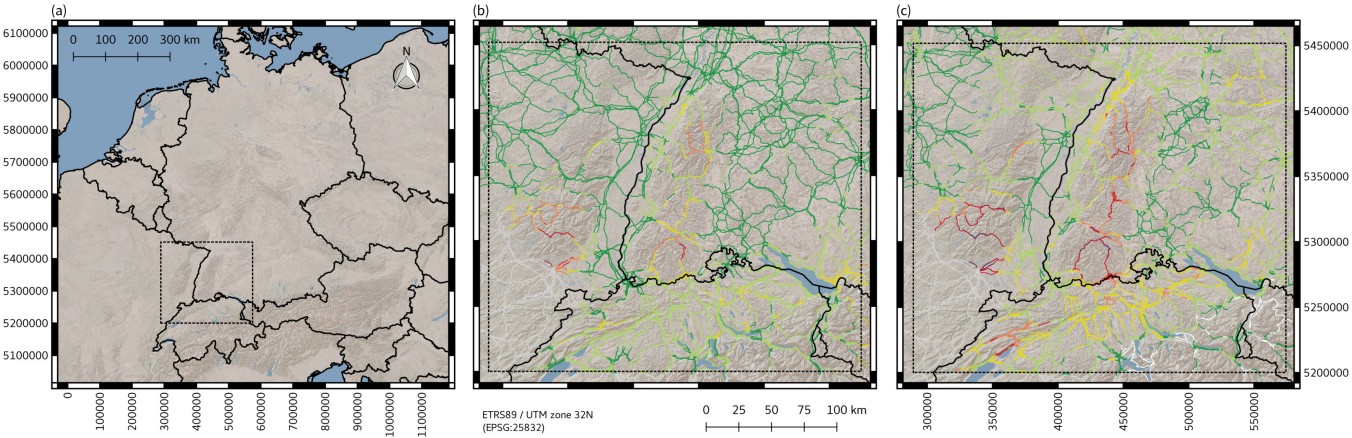

**Figure 3.** Overview of the target region: (a) Location of the region in Central Europe, median of the increase in landslide triggering climate events for (b) the near future and (c) the remote future.

As far as topography is concerned, results show that the target region is not only prone to an increased amount of rainfall 10    that induces landslides, but also susceptible to mass wasting due to its topographic properties (Fig. 4). Both road and rail infrastructure in this area are frequently located in rugged terrain, in valleys, on hillsides or at foothills of mountains. This is particularly the case for the Rhine-Alpine Core-Network Corridor, parts of which are located along the steep western slopes of the Black Forest.

Regarding nature and the properties of the ground (Fig. 5) sedimentary rock types (sandstone, mudstone and limestone) 15    are prevalent in the area. Igneous (granite and basalt) and metamorphic (gneiss and schists) rocks can be found in the Vosges and Schwarzwald mountain ranges as well as the Kaiserstuhl volcano. Generally speaking, the higher the rock strength, the more rainfall is required to trigger landslides. It has been found that the rainfall threshold for sandstone and marl is smaller than the threshold for limestone, with limited marl and chert, which in turn is lower than the threshold for metamorphic rocks (Peruccacci et al., 2017). Cambisols are the prevailing soil type across major parts of the study area. The Upper Rhine Plane 20    shows a predominance of Fluvisols. While the Vosges as well as the Palatinate Forest are characterized by Podzols, the Swabian Alb is mainly covered by Leptosols. The South-Eastern parts, towards the alpine foothills, are highly structured – soil types

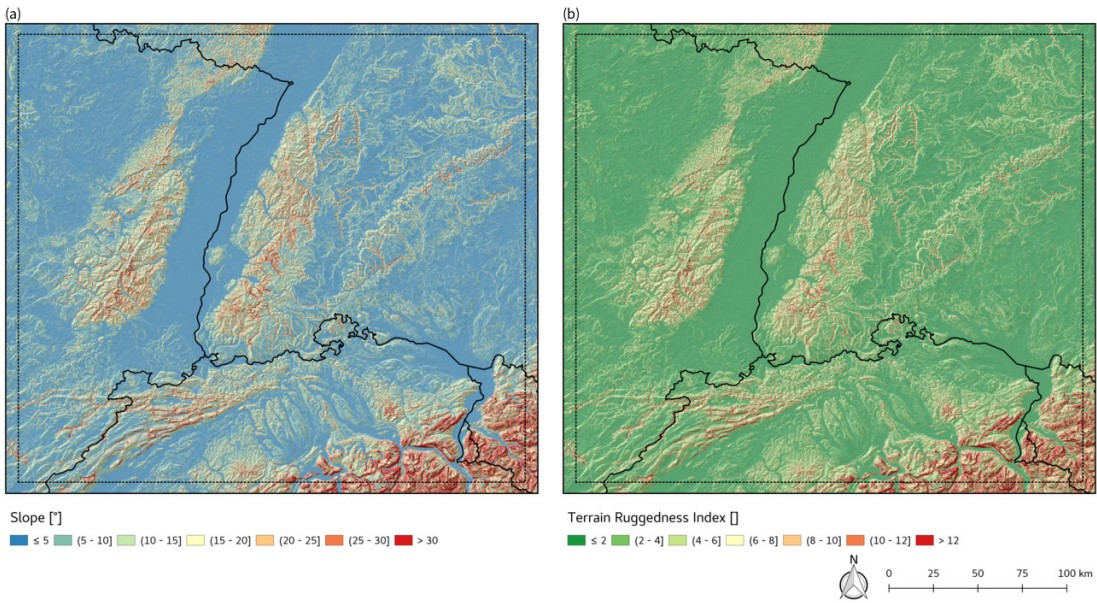

**Figure 4.** (a) Slope and (b) Terrain Ruggedness Index in the target region.

with a clay-enriched subsoil, soils influenced by water, as well as Leptosols and Podzols are common in these areas. Landslide susceptibility studies in the Swabian Alb have found that slopes with angles from 11° to 26°, consisting of colluvial layers, rendzic Leptosol and clayey soils superimposed on marl debris are indicators for slope instability in this area (Terhorst, 2007; Neuhäuser and Terhorst, 2007).

Our analysis is closed by mapping rainfall erosivity in terms of the R-factor and land cover for the target region (Fig. 6). R-factors between 700 and 1500 $\mathrm{MJ\,mm\,ha^{-1}\,h^{-1}\,yr^{-1}}$ can be observed across extensive parts of the target region. While rainfall erosivity is comparably low in the lowlands of the Alsace, it rapidly increases in the adjacent uplands, particularly towards the South. As pointed out by Panagos et al. (2015), a combination of large rainfall amounts and high erosivity densities – a condition that applies to major parts of the target region – is a precondition for landslides.

Concerning land cover, the target region is characterised by a multitude of small-scale areas featuring different land cover classes, which are typical for heterogeneous areas in complex terrain in Central Europe. Lowland areas are typically characterized by arable land (e.g. vineyards in the Upper Rhine Plain), while forests prevail with increasing altitude. It is broadly recognized in landslide research literature that forested areas favor terrain stability, while agricultural areas and abandoned cultivated lands that gradually recovered through natural vegetation are particularly susceptible to instability (Persichillo et al., 2017; Peruccacci et al., 2017; Petitta et al., 2015; Reichenbach et al., 2014; Beguería, 2006).

Consequences of traffic interruptions in this area are severe, as, for instance, the closure of the Rhine valley railway from August 13 to October 2, 2017 has shown. This interruption of this important north-south transport corridor, caused by mass movements at the construction site of the Rastatt-Tunnel, has led to considerable delays and increased costs, particularly for

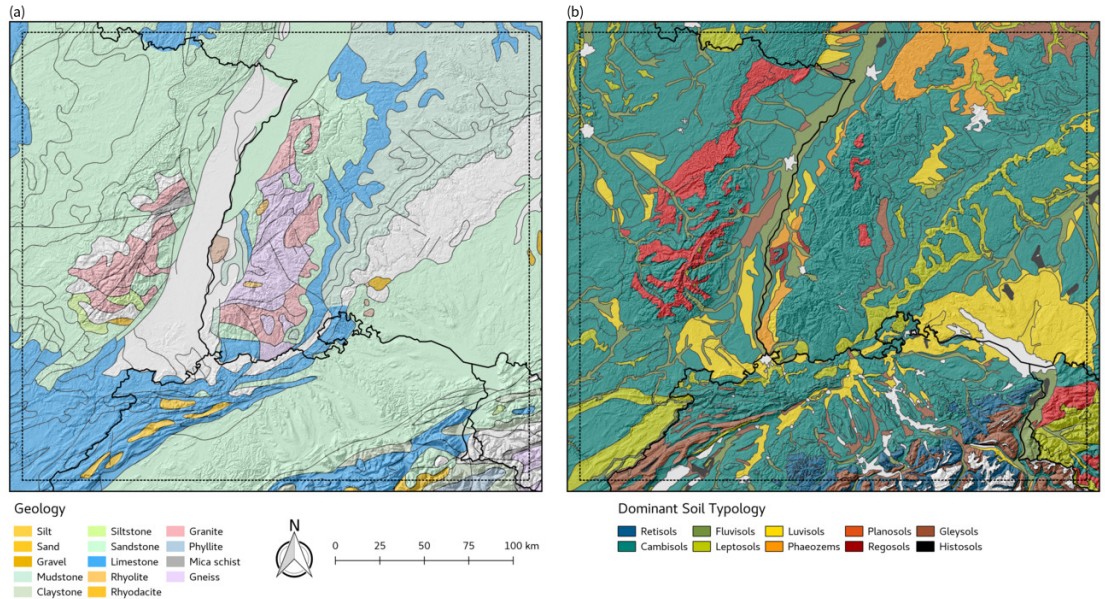

**Figure 5.** (a) Geology, including geological faults and (b) soil in the target region.

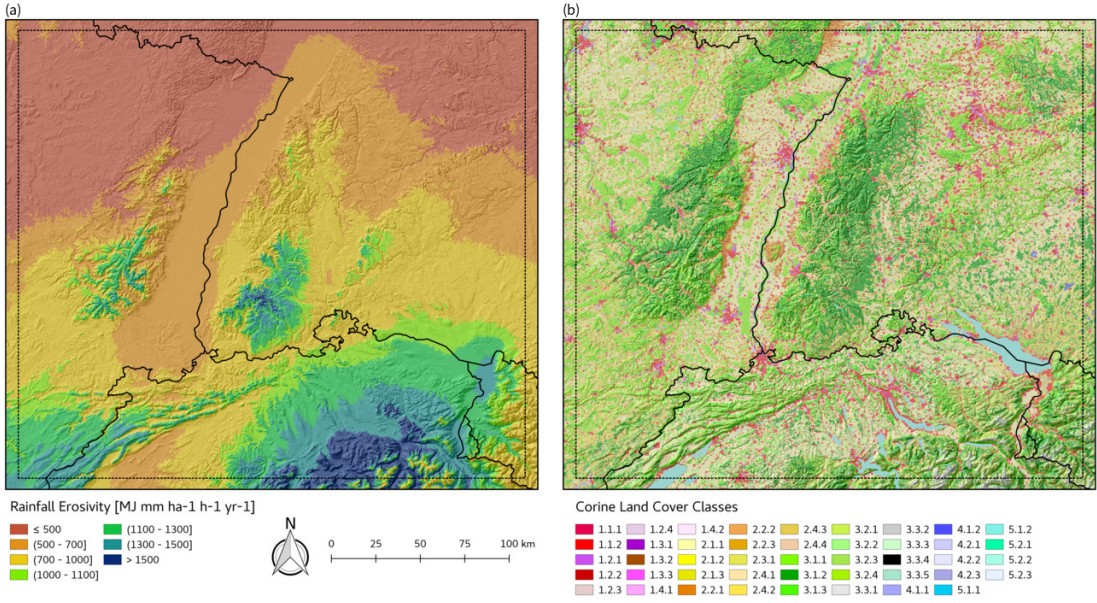

**Figure 6.** (a) Erosivity and (b) land cover in the target region.

freight transport. Usually, up to 350 trains use this traffic artery – which connects Italian ports with the ports of Rotterdam, Hamburg and Bremerhaven as well as Basel to the container port Duisburg – per day. The interruption of the track at Rastatt

has resulted in a massive congestion of freight trains all along the route due to the lack of alternative transport lines as well as the lack of engines and railroad engineers at alternative routes (FAZ, 2017; Ackeret, 2017; Gafner and Sommer, 2017). In addition to the direct economic damage, indirect and intangible costs such as noise disturbance as well as air pollution caused by an increased amount of cargo trains and heavy goods vehicles at alternative routes or an increased travel time for commuters, travellers and vacationers have to be considered (Postance et al., 2017).

## 5 Outlook

In this paper we highlight potential future changes of rainfall-related climate phenomena linked to landslide activities. We derive and present general patterns regarding the future rainfall-induced landslide exposure of road and railway networks in Central Europe and delineate the enrichment of such climate change related exposure information with additional relevant geodata in selected risk-prone areas.

Our findings indicate overall increases of landslide activity in the Central European region. However, while results suggest that flat orography at low altitudes will face only minor increases of potentially landslide-inducing rainfall periods per year until the end of this century, this effect is far more severe in complex terrain, where increases of up to 14 additional periods have to be expected. This general spatial distribution emerges already in the near future (2021-2050) but gets more pronounced in the remote future (2071-2100).

Most notably, mountainous regions and rolling landscapes – which exhibit the strongest increases in landslide-inducing rainfall periods – are generally particularly susceptible to landslides due to their topographic properties. Against the background that land bound transport infrastructure networks in higher elevated regions usually do not feature redundant elements at comparable economic efficiency, an increased exposure towards landslide-inducing rainfall events is likely to have potentially severe impacts on alpine communities if appropriate adaptation measures and precautions are neglected.

On the example of a particularly risk prone area covering parts of France, Germany, Switzerland and Austria, the added value attained by including additional maps derived from various geodata sources is demonstrated. Considering these regional conditions is of core importance for putting the mere exposure to climate events in a more context.

Future work should focus on the implications of exposure analyses for infrastructure users. This includes the extension of the investigated region to the whole of Europe as well as the consideration of additional climate indices describing further climate driven impacts. In order to provide decision support for stakeholders, future efforts should include network analysis and traffic modelling, as this will foster a deeper insight into regional mobility behaviour and criticality assessment of important links in the network. Eventually, investigating socio-economic impacts as well as vulnerability and risk management is of prime importance for the establishment of effective adaption measures in the context of climate change.

*Data availability.* Strong focus was put on using open access data wherever possible. Road data are accessible via the OSM API under http://api.openstreetmap.org/ and railroads data from Natural Earth are available at https://github.com/nvkelso/natural-earth-vector/tree/

master/10m_cultural or via the frontend at http://www.naturalearthdata.com/downloads/10m-cultural-vectors/railroads/. CORINE Land Cover data and the EU-DEM are provided by the Copernicus land monitoring service and can be accessed at http://land.copernicus.eu/. The geology data set (IGME 5000) is provided by the Federal Institute for Geosciences and Natural Resources at https://produktcenter.bgr.de/. Soil data

5 and rainfall erosivity are accessible through the European Soil Data Centre (ESDAC) at http://esdac.jrc.ec.europa.eu/resource-type/datasets.

*Competing interests.* The authors declare that they have no conflict of interest.

*Acknowledgements.* The authors would like to thank Melitta Dragaschnig (AIT) for providing assistance with the OSM data extract. We thank the two anonymous reviewers as well as the editor for providing insightful comments and suggestions on an earlier draft of the manuscript.

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
