# Peer review of "Potential future exposure of European land transport infrastructure to rainfall-induced landslides throughout the 21st century"

_Natural Hazards and Earth System Sciences, 2017_

## Referee Comment (RC1) · Anonymous Referee #1 · 12 Dec 2017

This paper displays some useful and interesting results and will be a useful paper to publish. It contributes to the understanding of landside hazards and provides new ideas. There are a number of issues with the methods section, as the results presented are not described by the methods. In order for this paper to be duplicable, the methods section needs to be completely re-written.

Scientific Significance: The paper will potentially offer a contribution to the understanding of landslide hazards and provides some new ideas.

[Figure]

Scientific Quality: The approaches used are confused, and methods are not apparent. The results and discussion section is unclear, and contains datasets and ideas which are not previously presented in either the introduction or methods sections (e.g. the CORINE dataset is first mentioned on p. 9 in the results and discussion section). References are generally appropriate, although there is an overreliance on IPCC publications, and there are a number of recent publications I consider to be missing.

Presentation Quality: The results and discussion are not presented in a clear and concise manner. This is due to the methods section not describing the methods used. Structure is therefore lacking as much of the results and discussion section brings in new ideas and analyses not previously discussed. Language used is not always appropriate and there are grammatical and spelling errors.

Specific comments:

P. 3, L. 23 – The authors select a threshold of 37.3mm and 25.6mm however, references for these are not introduced until p. 4.

P. 4 – The methods section does not describe the results discussed in the results and discussion section. This section needs to be re-written to ensure the duplicability of the study.

P. 4, L. 29 – KLIWAS17 is not introduced at all.

P. 5, L. 5/6 – "While the first row of each figure refers to the near future, the second row displays projection results for the remote future. The three columns represent the quartiles in increasing order respectively" should be in the figure caption, not the text.

P. 5/6 – Figures 1 and 2 are unclear. The authors need to clarify what the data are and how these were calculated.

P. 6/7 – It is unclear whether the authors are talking about landslide events or precipitation events when discussing change between the two reference periods.

P. 8 – I am not sure how the authors use topography and lithology/geology or soil typology in their analysis.

P. 8/9 – Figures 3 and 4 are good, but again, I am not sure how these were calculated; the authors need to clarify this.

P. 9. I am not sure how the authors use erosivity or land cover in their analysis.

In the abstract, the authors state "Results indicate overall increases of landslide occurrences. While flat terrain at low altitudes exhibits increases of about two more landslide events per year until the end of this century, higher elevated regions are more affected and show increases of up to eight additional events", but do not show how these results are obtained from descriptions provided in the methods section. Furthermore, this "result" is not mentioned in the results and discussion section and also not mentioned in the conclusion; this needs to be clarified.

P. 10 – I am not sure how the mapping for Figures 5 and 6 was carried out. The authors need to be clear when it is their work that is being referenced, or the work of others. Figures must be referenced correctly if data were obtained from other sources. The use of these data must also be described in the methods section as these are first introduced in the results and discussion.

P. 11 – I think that the authors conclusion sums up the findings of the paper, but does not reflect the results and findings described in the abstract and is confusing to read for this reason.

---

## Short Comment (SC1) · 12 Dec 2017

I would like to thank the referee for the evaluation of our manuscript and the provided feedback.

Thank you for pointing out possible difficulties in understanding parts of the results and discussion section without a proper introduction of certain data sets used and a more concise description of the methods applied. This is helpful information, since criticism in this respect is indeed justified. While data sets used are referenced at the

end of the manuscript in the *Data availability* section, not all of them are described in the respective sections in the text. We will rewrite and extend the data and methods sections accordingly. Clarification in this respect should straighten out most of the issues raised by the reviewer.

---

## Referee Comment (RC2) · Anonymous Referee #2 · 15 Dec 2017

GENERAL COMMENT

The article presents an evaluation of possible future variations in the overcoming of an already-defined rainfall threshold for landslide occurrence in Central Europe, as a result of the application of an ensemble of downscaled climate projections, with particular regard to roads and railways.

The paper is clear, sufficiently well-written and potentially publishable. It follows somehow the IMRaD structure, even if with some drawbacks, that should be improved. The

[Figure]

English language is good. In my opinion, the manuscript needs major revisions before being accepted for publication, for several reasons listed below.

Mainly, the theoretical background and the proposed method are not well defined. In particular, the definition of the climate index is not well explained in the text (it can be deduced after reading the results), and the procedure for obtaining the maps of changes in threshold exceedance related to infrastructures are not clear.

Moreover, is not explained how the Authors used the information contained in the maps of slope, TRI, geology, soil types, rainfall erosivity, and CLC. These maps were used only for comparison with the obtained results? Or they were used also in the calculations? This should be explained.

In several parts of the manuscript, Authors refer both to "landslides" and "landslide events". I would suggest to define what a "landslide event" is or to use the simple "landslide".

If I have well understood, Authors are referring on threshold overcoming as climate index for landslide occurrence. That's true? If yes, this should be reported and defined clearly in the text.

Moreover, several toponyms and names of regions are reported in the text. A map with all those names (also as supplementary material) would be useful for non-Europeans readers. At least, I suggest adding the names of the Countries in which the cited regions are located (e.g., Alsace, France).

For what concern the structure of the paper, the "Introduction" section is not very easy-to-read. At the beginning there is a summary of the work (lines 3-6), which should be better located and the end of the section.

The "Data" section is good, but another subsection with details about other used data could be appreciated.

The "Method" section is not clear. The definition of the climate index is not effective and

the procedure used to pass from whole maps to infrastructure maps should be better described. Moreover, the cited work made by Matulla et al. (2017) present a very similar approach and similar results. Thus, differences and improvements proposed in this new paper should be strongly described.

The "Results and Discussion" section is well-structured. However, two subsection could be added, referring to "Central Europe" and "Target area".

The "Outlook" section should be reworded, presenting the main findings and innovations of the work, and not only the future developments.

The reference list is complete, and all the articles are cited in the text.

SPECIFIC COMMENTS

The abstract should be shortened, particularly in the parts at lines 1-8 and 20-25.

The Central European region should be geographically defined.

Please, for a good understanding, add letters (a, b, . . .) in the panels of all figures. Consequently, add information in the captions. As an example, the text reported at page 5, lines 5-6 (The first row of each figure refers to the near future, the second row displays projection results for the remote future. The three columns represent the quartiles in increasing order respectively) should be better written in the caption of Figure 1. The same for the other figures.

Table 1 could be changed in two tables: one referring to values related to roads, and another with values related to railways. In the caption, just mention the CI, without repeating the threshold values. I suggest moving to the method section the text reported at page 8 lines 6-9 and lines 15-20. Figure 6b: I suggest considering only the second level of the CLC classification. Please be sure that all the characters in the figures will be readable.

Please check the text and correct some typos and errors in referencing in the text.

[Figure]

Finally, I would suggest some works dealing with: i) climate change and infrastructures (Loveridge et al., 2010); ii) landslide hazard and risk hotspots in Europe (Jaedicke et al., 2014); iii) effects of environmental changes on landslide occurrence (Begueria, 2006; Gariano et al., 2017), on susceptibility evaluations (Van Den Eeckhaut et al., 2012; Pisano et al., 2017), and on risk analysis (Papathoma-Köhle and Glade, 2013; Promper et al., 2014).

TECHNICAL CORRECTIONS

Page 2, line 11: note that "IPCC, 2014b" is cited in the text before "IPCC, 2014a".

Page 2, line 20: add a space in "Europe(Nestroy".

Page 2, line 26: correct GCMs.

Page 3, line 3: delete comma after "both".

Page 3, line 4: correct brackets in "key determinants affecting landslides Gariano and Guzzetti (2016); Sidle and Ochiai (2013).".

Page 3, line 23: insert here the references for the thresholds that define the climate index.

Page 4, line 29: KLIWAS17 is not defined in the text.

Page 5, caption of figure 1: replace "&" with "and".

REFERENCES

Beguería, S.: Changes in land cover and shallow landslide activity: a case study in the Spanish Pyrenees, Geomorphology, 74(1–4), 196–206, doi:10.1016/j.geomorph.2005.07.018, 2006.

Gariano, S.L., Petrucci, O., Rianna, G., Santini, M., and Guzzetti, F.: Impacts of past and future land changes on landslides in southern Italy, Reg. Environ. Change, doi: 10.1007/s10113-017-1210-9, 2017.

Jaedicke, C., Van Den Eeckhaut, M., Nadim, F., Hervás, J., Kalsnes, B., Vangelsten, B.V., Smith, J.T., Tofani, V., Ciurean, R., Winter, M.G., Sverdrup-Thygeson, K., Syre, E., and Smebye, H.: Identification of landslide hazard and risk hotspots in Europe, Bull. Eng. Geol. Environ., 73(2), 325–339, doi:10.1007/s10064-013-0541-0, 2014.

Loveridge, F., Spink, T., O'Brien, T., Briggs, K., and Butcher, D.: The impact of climate and climate change on infrastructure slopes with particular reference to southern England, Q. J. Eng. Geol. Hydroge., 43(4), 461-472, doi:10.1144/1470-9236/09-050, 2010.

Papathoma-Köhle, M., and Glade, T.: The role of vegetation cover change for landslide hazard and risk, in: Renaud, F., Sudmeier-Rieux, K., Estrella, M. (eds.) The role of ecosystems in disaster risk reduction. UNU-Press, Tokyo, 293–320, 2013.

Pisano, L., Zumpano, V., Malek, Z., Rosskopf, C.M., and Parise, M.: Variations in the susceptibility to landslides, as a consequence of land cover changes: a look to the past, and another towards the future, Sci. Total Environ., doi:10.1016/j.scitotenv.2017.05.231, 2017.

Promper, C., Puissant, A., Malet, J.-P., and Glade, T.: Analysis of land cover changes in the past and the future as contribution to landslide risk scenarios, Appl. Geogr. 53, 11–19, doi:10.1016/j.apgeog.2014.05.020, 2014.

Van Den Eeckhaut, M., Hervás, J., Jaedicke, C., Malet, J.-P., Montanarella, L., and Nadim, F.: Statistical modelling of Europe-wide landslide susceptibility using limited landslide inventory data, Landslides, 9(3), 357–369, doi:10.1007/s10346-011-0299-z, 2012.

---

## Author Comment (AC1) · 8 Jan 2018

Matthias Schlögl and Christoph Matulla

matthias.schloegl@ait.ac.at

We would like to thank the referee for the evaluation of our manuscript and the provided feedback. Please find our responses below, with referee comments in italics, and authors' responses in standard format.

**1 GENERAL COMMENT**

*The article presents an evaluation of possible future variations in the overcoming of an already-defined rainfall threshold for landslide occurrence in Central Europe, as a result of the application of an ensemble of downscaled climate projections, with particular regard to roads and railways. The paper is clear, sufficiently well-written and potentially publishable. It follows somehow the IMRaD structure, even if with some drawbacks, that should be improved. The English language is good. In my opinion, the manuscript needs major revisions before being accepted for publication, for several reasons listed below.*

- *Mainly, the theoretical background and the proposed method are not well defined. In particular, the definition of the climate index is not well explained in the text (it can be deduced after reading the results), and the procedure for obtaining the maps of changes in threshold exceedance related to infrastructures are not clear.*

We would like to thank the reviewer for pointing this out. The methods section has indeed some room for improvement. We will rework this section in order to provide a more concise and clear description of the methods employed. In particular, we will describe the definition of the climate index as well as the procedure for obtaining the maps.

- *Moreover, is not explained how the Authors used the information contained in the maps of slope, TRI, geology, soil types, rainfall erosivity, and CLC. These maps were used only for comparison with the obtained results? Or they were used also in the calculations? This should be explained.*

The maps have been used to support the discussion of the implications of meteorological impacts imposed by the CI in a more practical/realistic context. They

were not used for the calculation of the CI or the results presented in maps 1 and 2. We will clarify this in the text.

- *In several parts of the manuscript, Authors refer both to "landslides" and "landslide events". I would suggest to define what a "landslide event" is or to use the simple "landslide".*

We will replace "landslide event" with "landslide" throughout the text in order to avoid confusion in this respect.

- *If I have well understood, Authors are referring on threshold overcoming as climate index for landslide occurrence. That's true? If yes, this should be reported and defined clearly in the text.*

Yes, this is correct. We will clarify this in the text.

- *Moreover, several toponyms and names of regions are reported in the text. A map with all those names (also as supplementary material) would be useful for non-Europeans readers. At least, I suggest adding the names of the Countries in which the cited regions are located (e.g., Alsace, France).*

Thank you for pointing this out. We will provide a supplementary map containing all toponyms as proposed by the reviewer.

- *For what concern the structure of the paper, the "Introduction" section is not very easy- to-read. At the beginning there is a summary of the work (lines 3-6), which should be better located and the end of the section.*

We will move the respective part to the end of the section. In addition, the introduction will be reworked in order to be better readable.

- *The "Data" section is good, but another subsection with details about other used data could be appreciated.*

We will add another subsection covering all other data sources used for figures 5 and 6.

- *The "Method" section is not clear. The definition of the climate index is not effective and the procedure used to pass from whole maps to infrastructure maps should be better described. Moreover, the cited work made by Matulla et al. (2017) present a very similar approach and similar results. Thus, differences and improvements proposed in this new paper should be strongly described.*

We will rework the methods section. We will add more precise information on the CI and the procedure of deriving the infrastructure maps from the gridded data sets. We will point out the improvements of this work compared to the work by Matulla et al., 2017.

- *The "Results and Discussion" section is well-structured. However, two subsection could be added, referring to "Central Europe" and "Target area".*

We will add these two subsections.

- *The "Outlook" section should be reworded, presenting the main findings and innovations of the work, and not only the future developments.*

We will rework the outlook section by including the main findings and innovations of the present work as proposed by the reviewer.

- *The reference list is complete, and all the articles are cited in the text.*

**2  SPECIFIC COMMENTS**

- *The abstract should be shortened, particularly in the parts at lines 1-8 and 20-25.*

We will shorten the abstract to be more concise, particularly the parts mentioned by the reviewer.

[Figure]

- *The Central European region should be geographically defined.*

  We will define the region at the beginning of the proposed subsection 4.1.

- *Please, for a good understanding, add letters (a, b, . . .) in the panels of all figures. Consequently, add information in the captions. As an example, the text reported at page 5, lines 5-6 (The first row of each figure refers to the near future, the second row displays projection results for the remote future. The three columns represent the quartiles in increasing order respectively) should be better written in the caption of Figure 1. The same for the other figures.*

  We will rework the figures and their captions accordingly.

- *Table 1 could be changed in two tables: one referring to values related to roads, and another with values related to railways. In the caption, just mention the CI, without re- peating the threshold values.* While table 1 could indeed be changed into two tables, differences in the resulting summary statistics for road and railway network deviate only to a minor extent. The underlying climatological raster data sets are the same and both road and railway networks are quite dense in Central Europe. We would therefore suggest to keep one table, since the additional information gained by splitting the table into two tables is negligible, as both tables will look very similar.

  We will adjust the caption accordingly.

- *I suggest moving to the method section the text reported at page 8 lines 6-9 and lines 15-20.*

  We will move the paragraphs as proposed by the reviewer.

- *Figure 6b: I suggest considering only the second level of the CLC classification. Please be sure that all the characters in the figures will be readable.*

  We have decided to use all three levels of CLC in order to provide a consistent

land cover image. We have used the official legend and color coding for CLC level 3. By using only level 2, we would need to define custom colors for each level, which will lead to inconsistencies with other CLC maps.

- *Please check the text and correct some typos and errors in referencing in the text.*
  We will double-check the whole text in this respect.

- *Finally, I would suggest some works dealing with: i) climate change and infrastructures (Loveridge et al., 2010); ii) landslide hazard and risk hotspots in Europe (Jaedicke et al., 2014); iii) effects of environmental changes on landslide occurrence (Begueria, 2006; Gariano et al., 2017), on susceptibility evaluations (Van Den Eeckhaut et al., 2012; Pisano et al., 2017), and on risk analysis (Papathoma-Köhle and Glade, 2013; Promper et al., 2014).*

  We will consider including these references where appropriate.

**3 TECHNICAL CORRECTIONS**

All technical corrections will be implemented/corrected as proposed by the reviewer.

---

## Author Comment (AC2) · 8 Jan 2018

We would like to thank the referee for the evaluation of our manuscript and the provided feedback. Please find our responses below, with referee comments in italics, and authors' responses in standard format.

**1 General Comments**

*This paper displays some useful and interesting results and will be a useful paper to publish. It contributes to the understanding of landside hazards and provides new ideas. There are a number of issues with the methods section, as the results presented are not described by the methods. In order for this paper to be duplicable, the methods section needs to be completely re-written.*

**1.1 Scientific Significance**

*Scientific Significance: The paper will potentially offer a contribution to the understanding of landslide hazards and provides some new ideas.*

**1.2 Scientific Quality**

*The approaches used are confused, and methods are not apparent. The results and discussion section is unclear, and contains datasets and ideas which are not previously presented in either the introduction or methods sections (e.g. the CORINE dataset is first mentioned on p. 9 in the results and discussion section). References are generally appropriate, although there is an overreliance on IPCC publications, and there are a number of recent publications I consider to be missing.*

Thank you for pointing out possible difficulties in understanding parts of the results and discussion section without a proper introduction of certain data sets used and a more concise description of the methods applied. This is helpful information, since data and methods sections indeed offer room for improvement. While data sets used are referenced at the end of the manuscript in the data availability section, not all of them are described in the respective sections in the text. We will rewrite and extend the data and methods sections accordingly.
**1.3 Presentation Quality**

*The results and discussion are not presented in a clear and concise manner. This is due to the methods section not describing the methods used. Structure is therefore lacking as much of the results and discussion section brings in new ideas and analyses not previously discussed. Language used is not always appropriate and there are grammatical and spelling errors.*

Clarification of data and methods sections should straighten out these issues. In addition, we will rework the results section to match the reworked methods section.

**2 Specific comments**

- *P. 3, L. 23 – The authors select a threshold of 37.3 mm and 25.6 mm however, references for these are not introduced until p. 4.*

  We will add the reference to the CI on p3 l23.

- *P. 4 – The methods section does not describe the results discussed in the results and discussion section. This section needs to be re-written to ensure the duplicability of the study.*

  Thank you for pointing this out. We will rework the data and methods sections.

- *P. 4, L. 29 – KLIWAS17 is not introduced at all.*

  We will replace KLIWAS17 with "ensemble of 17 climate model runs (Imbery et al.,2013)" to avoid the introduction of a new term that is not used elsewhere in the manuscript.

- *P. 5, L. 5/6 – "While the first row of each figure refers to the near future, the second row displays projection results for the remote future. The three columns*
*represent the quartiles in increasing order respectively" should be in the figure caption, not the text.*

- *P. 5/6 – Figures 1 and 2 are unclear. The authors need to clarify what the data are and how these were calculated.*

  We will clarify this in the text, in particular by reworking the data and methods sections.

- *P. 6/7 – It is unclear whether the authors are talking about landslide events or precipitation events when discussing change between the two reference periods.*

  These two are linked to each other via the CI. Changes between the two reference periods refer to the changes in the CI. The CI is based on precipitation data, but serves as a proxy for landslide events. We will clarify this in the text.

- *P. 8 – I am not sure how the authors use topography and lithology/geology or soil typology in their analysis.*

  We will clarify this in the text, in particular by reworking the data and methods sections.

- *P. 8/9 – Figures 3 and 4 are good, but again, I am not sure how these were calculated; the authors need to clarify this.*

  We will clarify this in the text, in particular by reworking the data and methods sections.

- *P. 9. I am not sure how the authors use erosivity or land cover in their analysis.*

  We will clarify this in the text, in particular by reworking the data and methods sections.

- *In the abstract, the authors state "Results indicate overall increases of landslide occurrences. While flat terrain at low altitudes exhibits increases of about two more landslide events per year until the end of this century, higher elevated regions are more affected and show increases of up to eight additional events", but do not show how these results are obtained from descriptions provided in the methods section. Furthermore, this "result" is not mentioned in the results and discussion section and also not mentioned in the conclusion; this needs to be clarified.*

We will clarify this in the text.

- *P. 10 – I am not sure how the mapping for Figures 5 and 6 was carried out. The authors need to be clear when it is their work that is being referenced, or the work of others. Figures must be referenced correctly if data were obtained from other sources. The use of these data must also be described in the methods section as these are first introduced in the results and discussion.*

We will clarify this in the text, in particular by reworking the data and methods sections.

- *P. 11 – I think that the authors conclusion sums up the findings of the paper, but does not reflect the results and findings described in the abstract and is confusing to read for this reason.*

We will clarify this in the text and increase consistency with respect to conclusions throughout the manuscript.

---

## Author Response (AR1)

**Potential future exposure of European land transport infrastructure to rainfall-induced landslides throughout the 21$^{\text{st}}$ century**

Matthias Schlögl and Christoph Matulla

January 22, 2018

**1  General Response**

We would like to thank the two anonymous referees as well as the editor for the evaluation of our manuscript and the helpful feedback they provided. Both reviewers indicated similar items to improve, in particular as far as clarity of methods and structure is concerned. We have implemented the changes and improvements as proposed by the reviewers, putting special emphasis on the consistency of the manuscript. We are confident that the reworked data and methods sections should lead to a better golden thread throughout the manuscript.

Please find our detailed responses below, with referee comments in italics, and authors responses in standard format.

In addition, we have added a latexdiff version as this supplement to highlight the changes between the two manuscripts.

**2  Response to Reviewer 1**

**2.1  General Comments**

*This paper displays some useful and interesting results and will be a useful paper to publish. It contributes to the understanding of landside hazards and provides new ideas. There are a number of issues with the methods section, as the results presented are not described by the methods. In order for this paper to be duplicable, the methods section needs to be completely re-written.*

**2.1.1  Scientific Significance**

*Scientific Significance: The paper will potentially offer a contribution to the understand- ing of landslide hazards and provides some new ideas.*

**2.1.2 Scientific Quality**

*The approaches used are confused, and methods are not apparent. The results and discussion section is unclear, and contains datasets and ideas which are not previously presented in either the introduction or methods sections (e.g. the CORINE dataset is first mentioned on p. 9 in the results and discussion section). References are generally appropriate, although there is an overreliance on IPCC pub- lications, and there are a number of recent publications I consider to be missing.*

Thank you for pointing out possible difficulties in understanding parts of the results and discussion section without a proper introduction of certain data sets used and a more concise description of the methods applied. This is helpful information, since data and methods sections indeed offer room for improvement. While data sets used are referenced at the end of the manuscript in the data availability section, not all of them are described in the respective sections in the text. We have rewritten and extended the data and methods sections accordingly.

**2.1.3 Presentation Quality**

*The results and discussion are not presented in a clear and concise manner. This is due to the methods section not describing the methods used. Structure is therefore lacking as much of the results and discussion section brings in new ideas and analyses not previously discussed. Language used is not always appropriate and there are grammatical and spelling errors.*

We reworked both data and methods sections to clarify and straighten out this issue. In addition, we have reworked the results section to match the reworked methods section.

**2.2 Specific comments**

- *P. 3, L. 23 – The authors select a threshold of 37.3mm and 25.6mm however, refer- ences for these are not introduced until p. 4.*

  We have add the reference to the CI.

- *P. 4 – The methods section does not describe the results discussed in the results and discussion section. This section needs to be re-written to ensure the duplicability of the study.*

  Thank you for pointing this out. We have reworked the data and methods sections.

- *P. 4, L. 29  KLIWAS17 is not introduced at all.*

  We have replaced "KLIWAS17" with "ensemble of 17 climate model runs (Imbery et al.,2013)" to avoid the introduction of a new term that is not used elsewhere in the manuscript.

- *P. 5, L. 5/6 – "While the first row of each figure refers to the near future, the second row displays projection results for the remote future. The three columns represent the quartiles in increasing order respectively" should be in the figure caption, not the text.*

  We have moved this sentence to the figure caption as proposed by the reviewer.

- *P. 5/6 – Figures 1 and 2 are unclear. The authors need to clarify what the data are and how these were calculated.*

  We have clarified this in the text, in particular by reworking the data and methods sections.

- *P. 6/7 – It is unclear whether the authors are talking about landslide events or precipi- tation events when discussing change between the two reference periods.*

  These two are linked to each other via the CI. Changes between the two reference periods refer to the changes in the CI. The CI is based on precipitation data, but serves as a proxy for landslide events. We have also clarified this in the text.

- *P. 8 – I am not sure how the authors use topography and lithology/geology or soil typology in their analysis.*

  We have clarified this in the text, in particular by reworking the data and methods sections.

- *P. 8/9 – Figures 3 and 4 are good, but again, I am not sure how these were calculated; the authors need to clarify this.*

  We have clarified this in the text, in particular by reworking the data and methods sections.

- *P. 9. I am not sure how the authors use erosivity or land cover in their analysis.*

  We have clarified this in the text, in particular by reworking the data and methods sections.

- *In the abstract, the authors state "Results indicate overall increases of landslide occur- rences. While flat terrain at low altitudes exhibits increases of about two more landslide events per year until the end of this century, higher elevated regions are more affected and show increases of up to eight additional events", but do not show how these results are obtained from descriptions provided in the methods section. Furthermore, this "result" is not mentioned in the results and discussion section and also not mentioned in the conclusion; this needs to be clarified.*

  We have clarified this in the text.

- *P. 10 – I am not sure how the mapping for Figures 5 and 6 was carried out. The authors need to be clear when it is their work that is being referenced, or the work of others. Figures must be referenced correctly if data were obtained from other sources. The use of these data must also be described in the methods section as these are first introduced in the results and discussion.*

  We have clarified this in the text, in particular by reworking the data and methods sections.

- *P. 11 – I think that the authors conclusion sums up the findings of the paper, but does not reflect the results and findings described in the abstract and is confusing to read for this reason.*

  We have clarified this in the text and increased consistency with respect to conclusions throughout the manuscript.

**3 Response to Reviever 2**

**3.1 GENERAL COMMENT**

*The article presents an evaluation of possible future variations in the overcoming of an already-defined rainfall threshold for landslide occurrence in Central Europe, as a result of the application of an ensemble of downscaled climate projections, with particular regard to roads and railways. The paper is clear, sufficiently well-written and potentially publishable. It follows somehow the IMRaD structure, even if with some drawbacks, that should be improved. The English language is good. In my opinion, the manuscript needs major revisions before being accepted for publication, for several reasons listed below.*

- *Mainly, the theoretical background and the proposed method are not well defined. In particular, the definition of the climate index is not well explained in the text (it can be deduced after reading the results), and the procedure for obtaining the maps of changes in threshold exceedance related to infrastructures are not clear.*
  The methods section has indeed some room for improvement. We have reworked this section in order to provide a more concise and clear description of the methods employed. In particular, we have described the definition of the climate index as well as the procedure for obtaining the maps.

- *Moreover, is not explained how the Authors used the information contained in the maps of slope, TRI, geology, soil types, rainfall erosivity, and CLC. These maps were used only for comparison with the obtained results? Or they were used also in the calculations? This should be explained.*
  The maps have been used to support the discussion of the implications of meteorological impacts imposed by the CI in a more practical/realistic

context. They were not used for the calculation of the CI or the results presented in maps 1 and 2. We have added this explicitly (two times) in order to clarify this in the text.

- *In several parts of the manuscript, Authors refer both to "landslides" and "landslide events". I would suggest to define what a "landslide event" is or to use the simple "landslide".*
  We have replace "landslide event" with "landslide" throughout the text in order to avoid confusion.

- *If I have well understood, Authors are referring on threshold overcoming as climate index for landslide occurrence. Thats true? If yes, this should be reported and defined clearly in the text.*
  Yes, this is correct. We have clarified this in the text.

- *Moreover, several toponyms and names of regions are reported in the text. A map with all those names (also as supplementary material) would be useful for non-Europeans readers. At least, I suggest adding the names of the Countries in which the cited regions are located (e.g., Alsace, France).*
  Thank you for pointing this out. We have provided a supplementary map containing all toponyms. We have added this map at the end of this response.

- *For what concern the structure of the paper, the "Introduction" section is not very easy- to-read. At the beginning there is a summary of the work (lines 3-6), which should be better located and the end of the section.*
  As proposed by the reviewer, we have moved the respective part to the end of the section. In addition, we have reworked and slightly extended the introduction in order to be better readable.

- *The "Data" section is good, but another subsection with details about other used data could be appreciated.*
  We have added another subsection (2.4 Additional geodata) covering all other data sources used for figures 5 and 6.

- *The "Method" section is not clear. The definition of the climate index is not effective and the procedure used to pass from whole maps to infrastructure maps should be better described. Moreover, the cited work made by Matulla et al. (2017) present a very similar approach and similar results. Thus, differences and improvements proposed in this new paper should be strongly described.*
  We have reworked the methods section. We have added more precise information on the CI and the procedure of deriving the infrastructure maps from the gridded data sets. We have pointed out the improvements of this work compared to the work by Matulla et al., 2017.

- *The "Results and Discussion" section is well-structured. However, two subsection could be added, referring to "Central Europe" and "Target area".*
  We have added these two subsections as proposed by the reviewer.

- *The "Outlook" section should be reworded, presenting the main findings and innovations of the work, and not only the future developments.*
  We have reworked the outlook section by including the main findings and innovations of the present work as proposed by the reviewer.

- *The reference list is complete, and all the articles are cited in the text.*

**3.2 SPECIFIC COMMENTS**

- *The abstract should be shortened, particularly in the parts at lines 1-8 and 20-25.*
  We have shortened the abstract to be more concise, particularly the parts mentioned by the reviewer.

- *The Central European region should be geographically defined.*
  We have defined the region in the introduction after the first occurrence of the term "Central Europe".

- *Please, for a good understanding, add letters (a, b, . . .) in the panels of all figures. Consequently, add information in the captions. As an example, the text reported at page 5, lines 5-6 (The first row of each figure refers to the near future, the second row displays projection results for the remote future. The three columns represent the quartiles in increasing order respectively) should be better written in the caption of Figure 1. The same for the other figures.*
  We have reworked the caption by moving the text mentioned by the reviewer from the text body to the caption. We have added letters to the figures as proposed and added them to the caption.

- *Table 1 could be changed in two tables: one referring to values related to roads, and another with values related to railways. In the caption, just mention the CI, without re- peating the threshold values.* While table 1 could indeed be changed into two tables, differences in the resulting summary statistics for road and railway network deviate only to a minor extent. The summary statistics are based on all raster cells that cover road or railway infrastructure segments. Since the underlying climatological raster data sets are the same and both road and railway networks are quite dense in Central Europe, the raster cells selected for calculating the summaries are almost identical. We would therefore suggest to keep one table, since the additional information gained by splitting the table into two tables is negligible.
  We have adjusted the caption accordingly to be clear on what the table describes.

- *I suggest moving to the method section the text reported at page 8 lines 6-9 and lines 15-20.*
  We have moved the paragraphs as proposed by the reviewer.

- *Figure 6b: I suggest considering only the second level of the CLC classification. Please be sure that all the characters in the figures will be readable.*
  We have decided to use all three levels of CLC in order to provide a consistent land cover image. We have used the official legend and color coding for CLC level 3. By using only level 2, we would need to define custom colors for each level, which will lead to inconsistencies with other CLC maps.

- *Please check the text and correct some typos and errors in referencing in the text.*
  We have double-checked the whole text in this respect.

- *Finally, I would suggest some works dealing with: i) climate change and infrastructures (Loveridge et al., 2010); ii) landslide hazard and risk hotspots in Europe (Jaedicke et al., 2014); iii) effects of environmental changes on landslide occurrence (Begueria, 2006; Gariano et al., 2017), on susceptibility evaluations (Van Den Eeckhaut et al., 2012; Pisano et al., 2017), and on risk analysis (Papathoma-Köhle and Glade, 2013; Promper et al., 2014).*
  We have consulted these works and included most of them as references into the manuscript.

**3.3   TECHNICAL CORRECTIONS**

All technical corrections have been implemented/corrected as proposed by the reviewer.

[Figure]

Figure 1: Supplement: Names and toponyms of Central Europe used in this manuscript

[revised manuscript text omitted]

---

## Author Response (AR2)

Dear Matthias and Christoph,

Many thanks for your thorough revisions to the paper. Both reviewers have commented on how the paper is significantly improved and with appropriate minor revisions, I will be happy to accept the paper.

*We would like to thank both the anonymous reviewers and the editor for their constructive feedback, which is much appreciated.*
* * *
**Reviewer 1** has requested one minor revision:
- a little typo regarding referencing at Page 5, lines 11-12 and at Page 16, lines 8-9.
  *We have corrected the typo in the bibtex file.*
* * *
**Reviewer 2** has identified a number of minor revisions, but also asked you to more broadly take a look at the length of paragraphs and some of the repetition in the methods section:

Overall, there has been a significant improvement since the first submission. The authors have done well to address the comments and rectify many of the issues and I would like to commend them on this.

- There are parts which I feel are quite repetitive; particularly across the data and methods sections.
  *We have tried to remove these repetitions, as highlighted in the attached diff file.*

- I think the paragraph structure is sometimes a little difficult to follow, with overly short paragraphs followed by much longer ones, including multiple discussion points; the authors should take a look at this (e.g. P10, lines 8-13 should be a paragraph, lines 13-P.11, a second paragraph).
  *We have reviewed the paragraph structure.*

- P.2, L.24 - References appear in wrong order
  *All references are sorted chronologically, and not alphabetically.*

- P.4, L.18-26 – This reads like part of the introduction and should be moved accordingly.
  *We have moved this part to the introduction as proposed by the reviewer.*

- P.4, L.30 - Remove the bracket
  *removed*

- P.5. - The authors discuss slope and TRI as being important in this analysis, but it is not clear if this has been included and how; this needs to be explicit. It is also well understood about the effects of geology and lithology, but again, I'm not sure how these data have been used.
  *In our point of view, this is motivated at p. 3, l.8-12; p5 l8-12 and very explicitly at p5. l23-24: "The data sets about topographic, geological and soil properties as well as data about rainfall erosivity and land cover data have solely been used to augment the discussion of the implications imposed by the derived CI."*

- P.9, L.7 - You introduce the concept of a "target area" in the results, but I think that this should be introduced in the methods as it's not imminently clear why or how the authors do this.
  *The target region was only implicitly mentioned in the methods section at p5, l.19. We have moved the description from the results to the methods section.*
  *The target region is described in the text as a "particularly risk-prone area" with respect to future changes in landslide-inducing rainfall events. We are under the impression that this choice seems to be reasonable by consulting Figs. 1, 2 and 3.*

- P.10, L.8 onwards - This section reads more like methods and should be moved and expanded as it provides insights into some of the other comments made.
  *We do not agree with the reviewer in this case. The passage describes the results of the mapping (as seen in Fig. 6), clearly referring to the properties of the target region. This description is consistent with the description of Figs.4 and 5.*

- I think the authors should summarise the findings of the research at the end of the discussion or in the outlook - I want to know how explicitly how landslides may change in the future. In the results, the authors demonstrate an increase in landslide interaction with road networks and transport, I think that this is important and should be made clear to the reader.

  We have added a more precise description to the outlook section as proposed by the reviewer.
* * *
I do not believe these revisions will be too time consuming to implement, and will improve the communication in the paper. Please do not hesitate to contact me if you have any questions.

Thank you to both of the anonymous reviewers for their input.

Best wishes,
Faith

[revised manuscript text omitted]